# Isolation of a Marker Olean-12-en-28-butanol Derivative from *Viscum continuum* E. Mey. Ex Sprague and the Evaluation of Its Antioxidant and Antimicrobial Potentials

**DOI:** 10.3390/plants13101382

**Published:** 2024-05-16

**Authors:** Sipho Mapfumari, Buang Matseke, Kokoette Bassey

**Affiliations:** Department of Pharmaceutical Sciences, School of Pharmacy, Sefako Makgatho Health Sciences University, Molotlegi Street, Ga-Rankuwa, Pretoria 0204, South Africa; sipho.mapfumaru@smu.ac (S.M.); buang.matseke@smu.ac.za (B.M.)

**Keywords:** mistletoe, isolation, olean-12-en-28-butanol, antioxidant, antimicrobial

## Abstract

*Viscum continuum* E. Mey. Ex Sprague (Mistletoe) is a semi-parasitic plant that grows on the branches of other trees with reported numerous biological activities. This study was aimed at isolating a compound/s that will be used as a standard reference for quality control of South African-based commercialized mistletoe products and to further perform antioxidant and antimicrobial tests on the isolated compound. A dried sample of mistletoe was ground and extracted successively with hexane, dichloromethane (DCM), acetone and methanol using a serial exhaustive cold maceration procedure. The compound was isolated using column chromatography, and its chemical structure was elucidated using two-dimensional nuclear magnetic resonance (2D NMR) and ultrahigh-performance liquid chromatography-mass spectrometry (UPLC-MS). The antioxidant activity of the compound was determined using DPPH, hydrogen radical scavenging activity and reducing power assays, whereas antimicrobial activity was assessed using the minimum inhibitory concentration (MIC) method. Subjection of the DCM extract to column chromatography resulted in the isolation of a compound elucidated as olean-12-en-28-butanol-1-one, 3-hydroxy-4,4,10, 14, 20-pentamethyl (**D4**). Both the DPPH, H_2_O_2_ radical scavenging activity and reducing power assays revealed a significant antioxidant potential of compound **D4** with an IC_50_ of 0.701 mg/mL, lower than that of gallic acid (0.793 mg/mL) for the H_2_O_2_ radical scavenging assay. The results also indicated good antibacterial activity of **D4** with an IC_50_ of 0.25 mg/mL, compared to ciprofloxacin with an IC_50_ of 0.0039 mg/mL, against two Gram-negative (*Pseudomonas aeruginosa*, *Escherichia coli*) and three Gram-positive (*Streptococcus pyogenes*, *Bacillus cereus and Staphylococcus aureus*) bacteria. This study is the first to report on the isolation of the olean-12-en-28-butanol derivative from mistletoe of the South African ecotype.

## 1. Introduction

Mistletoe has numerous biological activities, including antimicrobial and antioxidant activities, and the phytoconstituents in the plant are known to be responsible for its biological activities and such uses. The family of isolated compounds from mistletoe from other countries includes apoptosis-inducing and ribosome-inactivating lectins as well as cytotoxic thionins (viscotoxins) [1]. The other main compounds of *Viscum* L. species are flavonoids, phenolic acids, terpenoids, sterols, phenylpropanoids, and alkaloids [2]. Some compounds have been isolated from mistletoe, purified, and their structures duly elucidated. From the European white-berry mistletoe (*Viscum album* L.) that has found application in the treatment of cancer due to the presence of lectins, viscotoxins and alkaloids, novel amino alkaloids 4,5,4′-trihydroxy-3,3′-iminodibenzoic acid and 4,5,4′,5′-tetrahydroxy-3,3′-iminodibenzoic acid were isolated by [3] in 2012. Furthermore, 4-*O*-[β-D-Apiosyl(1→2)]-β-D-glucosyl]-5-hydroxyl-7-*O*-sinapylflavanone 3-(4-acetoxy-3,5-dimethoxy)-phenyl-2E-propenyl-β-D-glucopyranoside, 3-(4-hydroxy-3,5-dimethoxy)-phenyl-2E-propenyl-β-D-glucopyranoside, 5,7-dimethoxy-4-O-β-D-glucopyranoside flavanone, 4,5-dimethoxy-7-hydroxy flavanone, and 5,7-dimethoxy-4-hydroxy flavanone were isolated from the organic extracts of European mistletoe. These compounds exhibited potential for their anti-glycation and antioxidant activities [4].

As far as the African mistletoe is concerned, dihydoxylupeol palmitate in addition to three other compounds, namely 3-methoxy quercetin, 3,4,5-trimethoxy gallate, and friedelin, were isolated from the leaves of mistletoes species of the Nigerian ecotype [5]. A second research group from Nigeria also reportedly isolated and purified compounds including two new compounds—linamarin gallate and walsuraside B, together with nine known ones, namely catechin, epicatechin, epicatechin 3-*O*-gallate, epicatechin 3-*O*-(3-*O*-methyl) gallate, epicatechin 3-*O*-(3,5-*O*-dimethyl)gallate, epicatechin 3-*O*-(3,4,5-*O*-trimethyl)gallate, quercetin 3-*O*-*β*-d-glucopyranoside, rutin, and peltatoside, from the leafy twigs of Nigerian mistletoe. The linamarin gallate was characterized as an unusual cyanogenic glycoside, while compound epicatechin 3-*O*-(3,4,5-*O*-trimethyl) gallate was isolated for the first time from a natural source. The antioxidant activities of all of the isolated compounds were evaluated using the 2, 2-diphenyl-1-picrylhydrazyl (DPPH) assay [6]. Despite an extensive literature search, no compounds isolated from the South African mistletoe have been documented to date. As a result, the aim of this study was to isolate a compound from South African mistletoe extracts and to evaluate its antioxidant and antimicrobial potentials as a standard reference for quality control of South African mistletoe- based commercialized products.

## 2. Results

The masses and percentage yield of the mistletoe dry extracts were 13.13 g: 1.32% for the acetone, 19.04 g: 1.90% for the dichloromethane, 19.23 g: 1.92% for the hexane and 23.67 g: 2.37% for the methanol extracts respectively. A mass of 10.5 g of the methanol extract was used to attempt the isolation and purification of compounds using the adsorption and wet column packing procedure. The mobile phase used to elute the column was ACTN: EA: FA (5:2:1 *v*/*v*/*v*), and the test tube collected fraction ranged between 30 and 50 mm depending on the target compound bands of interest. A flow chart detailing the isolation of compounds from the dichloromethane and methanol extracts is depicted in Figure 1.

The isolation work was carried out on the two samples which demonstrated considerable antioxidant activity as well as anticancer potential obtained from a preliminary study. Those two extracts were DCM and methanol extracts. Therefore, to isolate compounds from the DCM extract, 5.10 g of the DCM extract was run on a column chromatograph using hexane: ethyl-acetate (9:1 *v*/*v*) as the mobile phase. Fractions 1 to 10 were pooled together, as they displayed a similar profile. This was then further purified using preparative TLC since the sample was small. This then resulted in sub-fraction **D1**. 

Column chromatography was performed a second time with another 5.02 g DCM extract using hexane: Ethyl-acetate: acetone (6:2:2 *v*/*v*/*v*) as the mobile phase. This resulted in 32 test tube (TT) fractions that were collected. The test tube fractions were bulked together due to the display of identical TLC profiles, i.e., having the same Rf values. Whereas TT1–TT24 were discarded because they showed zero compound bands even after characterization of the TLC plate, TT25 to TT32 were pooled together, as they appeared to have a similar profile and to a certain extent appeared semi-pure because they revealed a single spot on the TLC plate. After evaporating the pooled fractions, this sub-fraction was labeled **D2**. **D2** was recrystallized with methanol to yield pure sub-fraction **D3**. The residue from **D2** was further washed with DCM to afford sub-fraction **D4**.

### 2.1. Evaluation of the Purity of the Isolated Compound by TLC and UPLC-MS Analysis

To confirm if the single and compact spots from the TLC analysis translated to purity of the four isolates (**D1**–**D4**), 1.5 mg/mL solution of each isolate was analyzed using high-resolution UPLC with MS detector attached in negative and positive modes. Results of the analysis showed that **D1** and **D2** were not as pure, contrary to the TLC results. On the other hand, isolates **D3** and **D4** obtained by re-crystalizing **D2** using methanol and dichloromethane, respectively, as recrystallizing solvent indicated identical high-resolution mass to charge ratio (*m*/*z*) of 471.3496 [M + H] (S1) and with the highest purity of 96.2% especially when detected by photo diode array detector, which is considered good for a natural product isolate.

### 2.2. Elucidation of the Structure of D4 from Its 1D and 2D NMR Data

**D4** was isolated as a white solid, UV_CHCl3_ λ_max_ 226, 288 nm. The one-dimensional ^1^H NMR (600 MHz, Methanol-d4), ^13^C NMR (150 MHz, Methanol-d4), DEPT as well as the two-dimensional COSY, HSQC and HMBC of **D4** were interpreted with help of Mestre Nova^®^ and ACD labs structure elucidator. Chemical shifts, multiplicity and coupling constant are listed in Table 1. HRESIMS (positive ion mode) *m*/*z* 471.1241 [M + H]^−^ (Calcd for C_31_H_50_O_3 471.38_) was recorded for **D4**. The HRESIMS fragmentation pattern (S2) was *m/z* 471 (M + 1), 453 (M–C3_1_H_49_O_2_), 386 (M–C_27_H_27_O), 149 (M–C_11_H_17_), and 71 (M–C_4_H_7_O). An R_f_ value of 0.23 was also obtained for **D4** by TLC using Hex: EA: ACTN (6:2:2 *v*/*v*/*v*) as mobile phase. 

The C-13 spectra of **D3**–**D4** (S3) were overlaid against each other to further investigate the UPLC-MS data with regard to both having an identical mass-to-charge ratio. As evident in the spectra, both compounds afforded the same number of carbon signals as per their chemical shifts. However, there was a slight change at 183.14 for **D3** and 182.35 for **D4**. The slight chemical shift was thought to have occurred due to minor differences in the experimental conditions or human error. As a result, only the NMR data of D4 were then used to elucidate the structure of the compound. In terms of the proton NMR of **D4**, the number of protons (H) in **D4** integrated to 50, and their multiplicity and coupling constants are summarized in Table 1 below. 

The last one-dimensional experiment, the distortionless enhancement by polarization transfer (DEPT) of D4 was instrumental in indicating the number of CH_3_, CH_2_, and CH groups present in **D4**. From the experiment, the number of CH_3_, CH_2_ and CH groups were elucidated as 5, 12, and 5, respectively. Whereas the CH_3_ signaled at 1.01, 15.32, 15.53, 17.14, and 18.30 ppm, the CH_2_ signals appeared at 22.93, 23.40, 23.57, 25.93, 27.19, 27.69, 28.10, 33.81, 33.06, 32.63, 32.45, and 30.67 ppm, and those for the CH methylene group pitched further downfield at 37.10, 33.81, 33.06, 32.63, and 32.45 to suggest that **D4** is characterized by a large, saturated moiety. The carbons that did not appear in these experiments are usually assumed to be the quaternary carbons that are usually devoid of protons.

The importance of two-dimensional experiments in the structural elucidation of an unknown compound can never be overemphasized. In that regard, the HSQC NMR of D4 was investigated to unravel the presence of protonated and quaternary carbons that are present in the structure of **D4**, and HSQC experiments of **D4** were conducted. The results obtained suggested that **D4** comprises six quaternary carbons at 183.14 for a possible carboxylic group, 143.59, peculiar for an olefinic moiety, and 46.12, 39.20, 38.71, and 37.0, diagnostic for the potential presence of angular saturated groups in the structure of **D4**, as well as 24 protonated carbons. In addition to the protonated and quaternary carbons, there was a one ring system-substituted hydroxyl group that signaled at 79.04 ppm. This was followed by the HMBC NMR of **D4**. This experiment is indispensable when it comes to identifying the different moieties that make up the structure of an unknown molecule. As for **D4**, long 3*J* coupling constant correlations that translate to connectivity of the different moieties or fragments of its structure existed for protons and the corresponding carbons. The skeleton of **D4** (Figure 2.I) and its HMBC correlations between the protons and its carbons are depicted in Figure 2.II. Of immense importance are the connection between ring A and B at H-25 at 1.70 ppm and the C-10 bridge of both rings. The other vital HMBC connection occurred between the methyl H-26 at 5.22 ppm and C-13 at 143. 5 ppm to reveal the connection between the C ring and D ring system. The H-12 at 5.22 ppm connected to the C-13 at 143.5 and C-18 at 47.6 ppm, thus connecting the ring C to rings D and E. Finally, the HMBC experiment showed that the entire pentacyclic terpene system consisting of rings A–E was connected to the keto butanol unit of the structure through the long 3*J* correlation between H-22 protons at 1.92 and 1.90 ppm and the C-28 at 183.1 ppm to afford the full structure of **D4** in Figure 3A.

### 2.3. In Vitro Quantitative Antimicrobial (MIC) Analysis of D4 Isolated from South African Mistletoe Extract

The antibacterial activity of the **D4** against two Gram-negative (*Pseudomonas aeruginosa*, *Escherichia coli*) and three Gram-positive (*Streptococcus pyogenes*, *Bacillus cereus* and S. aureus) bacteria was undertaken, and the results are displayed in Table 2. We postulate that the identical MIC value of 0.25 mg/mL for all of the test pathogens, was due to bacterial resistance to the **D4**. This is because both Gram-negative and Gram-positive bacteria are known to be surrounded by a thick layer of peptidoglycan, but the layer is thicker in the Gram-negative species [7]. In addition, the presence of lipopolysaccharide, efflux pumps, and other components of the cell wall structure may contribute to resistance to antimicrobials [8]. Furthermore, both the Gram-negative and Gram-positive bacteria must have been susceptible to the treatment with **D4**, thus allowing easy diffusion into the bacterial cell wall. This may be attributed to the ability of **D4** to weaken the cell membrane permeability, channel through the proteins and/or collapse the cell wall of the test bacteria. Although the outer membranes vary amongst pathogens (Gram-negative bacteria), **D4** gained entry at 0.25 mg/mL and would have collapsed the peptidoglycan cross-linkage of the Gram-positive bacterial species, thus indicating that **D4** has a broad spectrum of activity. However, further investigation of the potential mechanism of action of **D4**, other than the inhibition of cell growth, may be considered. 

### 2.4. In Vitro Antioxidant Potentials of D4

#### 2.4.1. DPPH Radical Scavenging Activity

DPPH is a known stable free radical and is commonly used to evaluate the antioxidant activity of natural extracts and pure compounds to act as free radical scavenger or hydrogen donors [9]. This assay investigated the free radical scavenging potentials of **D4** in comparison to two standards, gallic acid and butylated hydroxy toluene (BHT). The results obtained (Figure 4) spotlight **D4** as having concentration-dependent antiradical activity. 

#### 2.4.2. Hydrogen Peroxide Radical Scavenging Activity

As for the hydrogen peroxide radical scavenging activities of **D4**, the BHT revealed the best activity from 0.2 to 1.0 mg/mL (Figure 5). The gallic acid, on the other hand, yielded the second-best activity between 0.2 and 0.59 mg/mL. However, **D4** outperformed gallic acid with better antioxidant activity of 44% to 70% at a concentration of 0.6–1.0 mg/mL.

#### 2.4.3. Reducing Power Activity of D4

The results, as depicted in Figure 6, indicate that compound **D4** has reducing power ability. Antioxidants can act as reductones by donating a hydrogen atom to free radicals and thereby terminating the free radical chain reaction. In this case, compound **D4** was able to reduce potassium ferricyanide to potassium ferrocyanide, which then reacted with ferric chloride to form ferric–ferrous complex. Compound **D4** followed a trend like that of the standards gallic acid and BHT, where the reducing power activity increased as the concentration increased.

#### 2.4.4. Concentration Half Minimum (IC_50_) Potential of D4

The IC_50_ values of **D4** in comparison to standards gallic acid and butyl hydroxy toluene are summarized in Table 3. The IC_50_ values are representative of the minimum inhibitory antioxidant potentials of **D4**. Whereas the IC_50_ value of BHT was the best at concentrations of 0.072 to 0.422 mg/mL, that of gallica acid ranged between 0.175 and 0.793 mg/mL, and D4 indicated potentials at IC_50_ values of 0.398–0.701 mg/mL.

## 3. Discussion

After collating all the spectrometric and chromatographic data together, the proposed structure of **D4** was elucidated as olean-12-en-28-butanol-1-one, 3-hydroxy-4,4,10, 14, 20-pentamethyl (Figure 3A). On revising the stereochemistry of **D4** at positions 1 of its butanol moiety and at position 3, 4, 10, 14 and 20 of the triterpenoid skeleton, its full name was proposed as olean-12-en-28-butanol-β-one-3β-hydroxy-4,4-bis-β,10α, 14α, 20α-pentamethyl. This compound is a triterpenoid-type olean-12-enen derivative, the kind that has been highlighted as one of 250 compounds that have been identified or isolated from the genus *Viscum* in a review by [10] and co-workers in 2021. A detailed and succinct literature search indicated that **D4** is an analogue of olean-12-en-28-oic, 3-hydroxy-15,16-[1-methylided) bisoxy] (Figure 3B), which we detected in a preliminary study by GCXGC-MS analysis as a major marker compound present in the dichloromethane extract of South African mistletoe extract. In addition, compounds **D4** have been reported from other plant species by [11] and from numerous *Viscum* species from other countries by authors including [12,13,14].

According to Gibbons [15], 0.064 mg/mL MIC for a natural product should be considered significant to mitigate pathogenic effects on host cells. As summarized in Table 2, **D4** exhibited good [8] to significant [9] antimicrobial activities for five test organisms (with MIC = 0.25 mg/mL) compared to a ciprofloxacin MIC = 0.0156–0039 mg/mL for the same pathogens.

There was direct proportionality in activity as the concentration of **D4** increased from 0.2 to 1.0 mg/mL in the DPPH free radical scavenging assay. Even though gallic acid and BHT exhibited the best radical scavenging of 98% at a concentration of 0.8–1.0 mg/mL (Figure 4), **D4** competed and indicated good antioxidant potential with these standards of almost 80% in the same concentration range. Whereas a similar trend was observed in the Fe3+ reducing power assay (Figure 6), **D4** indicated better antioxidant potentials than gallic acid, but less than BHT in the same concentration range (Figure 5).

The better antioxidant activity of **D4** over gallic acid at greater than or equal to 0.6 mg/mL tends to suggest that **D4** can be synthesized commercially and formulated as an active antioxidant pharmaceutical ingredient from that optimum concentration.

At the highest concentration (1 mg/mL), gallic acid showed the highest reducing power ability of 96.13%, followed by compound **D4** at 95.02% and BHT with reducing power of 94.64%. These results are good, as compound **D4** exhibited a higher reducing power ability than the standard BHT.

The antioxidant activity of **D4** did not come as a surprise. This is because such activities have been reported for oleanolic acid and its derivatives [16]. The minimum inhibitory concertation half minimal (IC_50_) values of **D4** and the standards were calculated, and the results are summarized in Table 3. The IC_50_ values agreed with the free radical scavenging results. That is, BHT had the lowest value for DPPH, followed by gallic acid. But in the hydrogen peroxide assay, it was BHT that had the minimal IC_50_ value, followed by **D4** and a rather higher value for gallic acid. The trend implies that the lower the IC_50_ value, the better the antioxidant activity of the test samples. The results of this study agree with previous studies regarding extracts and isolated olean-12-ene derivatives of the genus *Viscum*, with well-documented antioxidant and antimicrobial potentials [17,18].

## 4. Materials and Methods

### 4.1. Plant Extraction

All solvents used in the plant extraction were purchased from Rochelle Chemicals and Lab Equipment Cc, Johannesburg, South Africa. 

The ground plant sample (1.05 kg) was extracted using NUVE shaking water bath (ZT10.ST 30, KK05F01, Akyurt, Turkey) at 100 RPM with 2.5 L of n-hexane, dichloromethane, acetone, and methanol in a serial exhaustive cold maceration procedure. This was performed in a 24 h cycle for each run. The extracts which resulted from the process were then filtered and concentrated using a Stuart rotary evaporator (Re400, Cole-Parmer Ltd. Stone, Staffordshire, UK) and were allowed to dry fully under a stream of air. The mass and percentage yield of the mistletoe dried extract were then determined using standard equations.

### 4.2. UPLC-MS and NMR Instrumentation Used for Structural Elucidation

Each of the isolated compounds was separately introduced by full-loop injection (1.0 μL) into a UPLC (Waters Acquity chromatographic system, Waters, Milford, MA, USA). The MS, UPLC and gradient system used are listed in Table 4.

### 4.3. Nuclear Magnetic Resonance (NMR)

Approximately 20 mg of each compound was dissolved in 2 mL of deuterated methanol (CD_3_OD) or chloroform (CDCl_3_), depending on the polarity, and transferred to an NMR tube prior to analysis. All spectra were recorded on a Bruker 600 Advance II NMR (Bruker, Billerica, MA, USA) at 600 MHz for 1H NMR and 150 MHz for ^13^C NMR at 25 °C, and the chemical shifts were recorded in parts per million (ppm). Two-dimensional (2D) NMR experiments were performed using standard Bruker microprograms. The solvent signals were used for calibration. Tetramethyl silane (TMS) was used as the reference solvent, and chemical shifts of the analyzed compound **D4** were recorded in parts per million. The raw free induction decay (FID) data obtained for the one-dimensional (^1^H, ^13^C) attached proton test (APT) and two-dimensional (COSY, HSQC and HMBC) experiments were analyzed and interpreted with the aid of Mestre nova^®^ (Mestre lab Research, S.L. Feliciano Barrera 9B-Bajo, 15706 Santiago de Compostela, Spain) and ACD/Lab’s structure elucidator (Canada).

### 4.4. Isolation from the Dichloromethane Extract

A mass of 5.02 g of the dichloromethane extract dissolved in 5.0 mL of dichloromethane was adsorbed to 12.5 g of dry silica (mesh 70–230) with the aid of mortar and pestle. The plant extract-silica mixture was allowed to air dry prior to loading on a column for separation. A 30 mm od × 2.0 mm wall × 600 mm long B24 socket and ground-glass stopper B24 with sintered disc P3 and PTFE S/C glass column (C.C. Immelmann (PTY) Ltd., Robertsham-Gauteng, South Africa) were mounted on a clamp support. The column was wet packed with a silica gel prepared by adding 50 g of dry silica in 50 mL of acetone. The silica slurry was filled up to 65% the length of the column. The residual solvent from the silica slurry was allowed to drip off the column until it was slightly above the packed silica gel. At this point, the dry silica-plant extract was loaded onto the wet silica gel. This was followed by the placement of cotton wool above the loaded plant extract. About 35% column length was allowed for the addition of the mobile phase. 

### 4.5. Biological Activity Assays of D4

#### 4.5.1. In Vitro Quantitative Antimicrobial (MIC) of D4

The isolated Gram-positive and -negative *P. aeruginosa* (ATCC9721), *S. pyrogenes* (ATCC19615), *E. coli* (ATCC105363)*, Bacillus cereus* (ATCC14579) and *S. aureus* (ATCC25923) strains were used because the minimum inhibitory concentration (MIC) values for **D4** and the Ciprofloxacin-positive control were determined based on a microbroth dilution method in 96 multi-well microtiter plates with slight modifications [19]. Each bacterial culture was prepared in Luria Berthani (LB) broth/agar and/or Mueller–Hinton broth (MHB). A 1.0 mg/mL solution of **D4** and standards was prepared. Briefly, 30 µL of Mueller–Hinton broth (MHB) was transferred into every well, and 50 µL of **D4** (in triplicate) was transferred into wells in Row A of the microtiter plate together with the negative (1% dimethyl sulfoxide) and positive (ciprofloxacin) controls. Additionally, a blank (sterile MH broth) and standardized bacterium (control) were prepared by transferring 50 µL to the wells, respectively. Two-fold serial dilutions were performed, resulting in decreasing concentrations over the range of 1000 to 1.0 μg/mL. Thereafter, 10 µL of the standardized bacterium was added into wells of the micro-well plate. After 24 h incubation at 37 °C, 10 μL of resazurin indicator solution (prepared by dissolving a 270 mg tablet in 40 mL of sterile distilled water) was added and incubated for a further 30 min to 1 h, until an optimal color developed. Bacterial growth inhibition (clear wells, no color change) was assessed visually and recorded. The MIC was recorded as the lowest concentration of the extract that inhibited bacterial growth.

#### 4.5.2. DPPH Free Radical Scavenging Activity of the Mistletoe Extracts

This was achieved following a method described in [20]. A variety of concentrations ranging from 0.2 mg/mL to 1.0 mg/mL were prepared for **D4**. Additionally, a DPPH solution was created with a concentration of 0.2 mg/mL. To test the **D4** antioxidant activity, 1.0 mL of the DPPH solution was mixed with 1.0 mL of **D4** solution in a test tube, and the contents were thoroughly mixed and vortexed before being placed in a dark cardboard box for 30 min. The spectrophotometric absorbance of the different concentrations was measured at 517 nm using a 96-well microplate-reader spectrophotometer (SprectraMax^®^, Molecular Devices, CA, USA). Gallic acid and butylated hydroxyl toluene (BHT) were used as reference standards at the same concentration. The percentage of radical scavenging activity of D4 was calculated using Equation (1):%DPPH radical scavenging activity = A_0_ − A_s_/A_0_ × 100(1)
where A_0_ is the absorbance of the negative control, and A_s_ is the absorbance of the extracts/ standards.

#### 4.5.3. Hydrogen Radical Scavenging Activity

The hydrogen peroxide scavenging potential of the mistletoe extracts was assessed using the method described in [20]. To conduct an experiment, a 2 mL solution of hydrogen peroxide (20 mM) was prepared in phosphate buffer saline with a pH of 7.40. To this solution, varying concentrations (ranging from 0.2 mg/mL to 1.0 mg/mL) of **D4** from stock solutions were added in increments of 1.0 mL. The resulting mixture was mixed thoroughly using a vortex and incubated for 10 min before measuring the absorbance at 560 nm using a spectrophotometer. To ensure accuracy, the reference standards used for this experiment were 1.0 mg/mL of both gallic acid and butylated hydroxyl toluene (BHT). The percentage of hydroxyl radical scavenging activity was calculated as follows:% Hydroxyl radical scavenging activity = A_0_ − A_s_/A_0_ × 100(2)
where A_0_ is the absorbance of the negative control, and A_s_ is the absorbance of the extracts/standards.

#### 4.5.4. Ferric Chloride Reducing Power Assay

The ferric chloride reducing power assay of samples was evaluated using the method in [20]. To start the experiment, **D4** was dissolved in dichloromethane. Then, a range of concentrations from 0.2 mg/mL to 1.0 mg/mL was prepared. Each concentration was mixed in a test tube with 2.5 mL of 0.2 M phosphate buffer (pH 6.6) and 2.5 mL of 1% (*w*/*v*) potassium ferricyanide (K3Fe (CN)6). The contents were mixed and incubated at 50 °C for 20 min. After this, 2.5 mL of 10% (*w*/*v*) trichloroacetic acid was added, and the mixture was centrifuged for 10 min at 3000 rpm. The upper layer of the resulting solution (2.5 mL) was mixed with 2.5 mL of distilled water and 0.5 mL of ferric chloride (0.1% *w*/*v*). The spectrophotometer was used to measure the absorbance of the resulting mixture at 700 nm. This procedure was repeated for the reference standards, gallic acid and butylated hydroxyl toluene. The percentage reducing power of **D4** was determined using Equation (3):% Reducing power = A_0_ − A_s_/A_0_ × 100(3)
where A_0_ is the absorbance of the negative control, and A_s_ is the absorbance of the extracts/ standards.

## 5. Conclusions

The current study successfully isolated the compound olean-12-en-28-butanol-β-one-3β-hydroxy-4,4-bis-β,10α, 14α, 20α-pentamethyl (**D4**) from a DCM extract of *Viscum continuum* E. Mey. Ex Sprague and evaluated its antioxidant and antimicrobial potentials. The study also proved that compound **D4** possesses antioxidant and antibacterial activity. All the quantitative methods used showed that the antioxidant activity of compound **D4** was concentration dependent. Antioxidants protect the body from oxidation, thereby reducing the risk of many chronic conditions such as cancer, diabetes, and heart attack. Because antioxidants protect against various conditions, these results support the tremendous traditional use of the plant throughout the world. The purpose of the study was achieved by isolating a bioactive compound from South African *Viscum continuum* E. Mey. Ex Sprague and evaluating its antioxidant and antimicrobial potentials.

## Figures and Tables

**Figure 1 plants-13-01382-f001:**
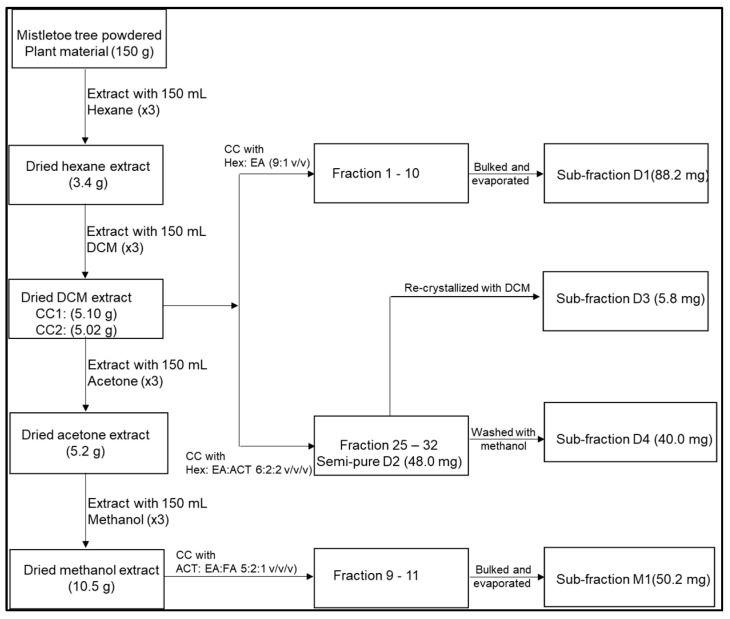
Flow chart detailing the isolation of compounds from the dichloromethane and methanol extracts of South African mistletoe.

**Figure 2 plants-13-01382-f002:**
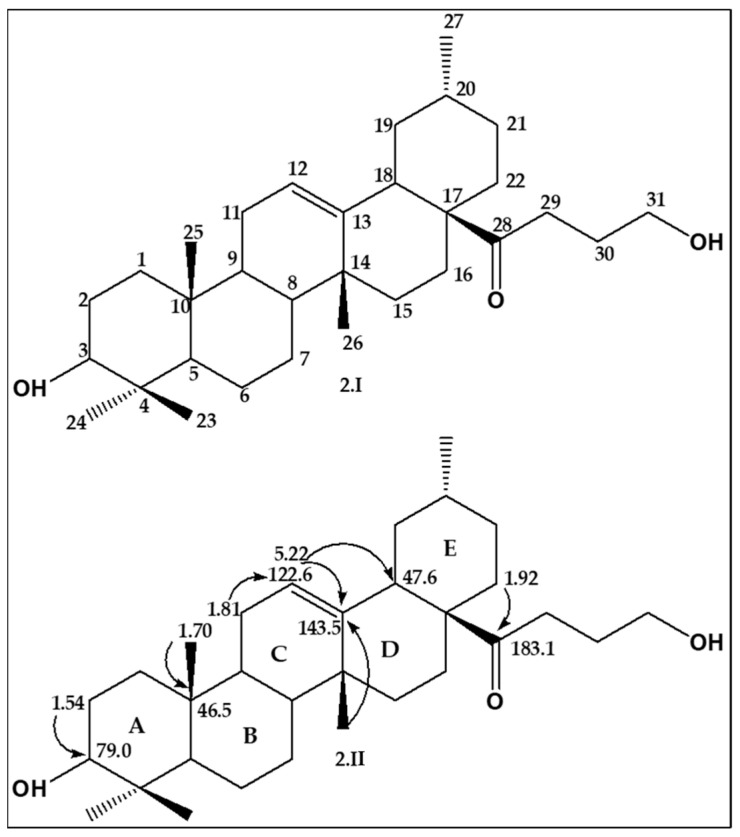
Carbon skeleton (2.I) and the HMBC correlations (2.II) of **D4**.

**Figure 3 plants-13-01382-f003:**
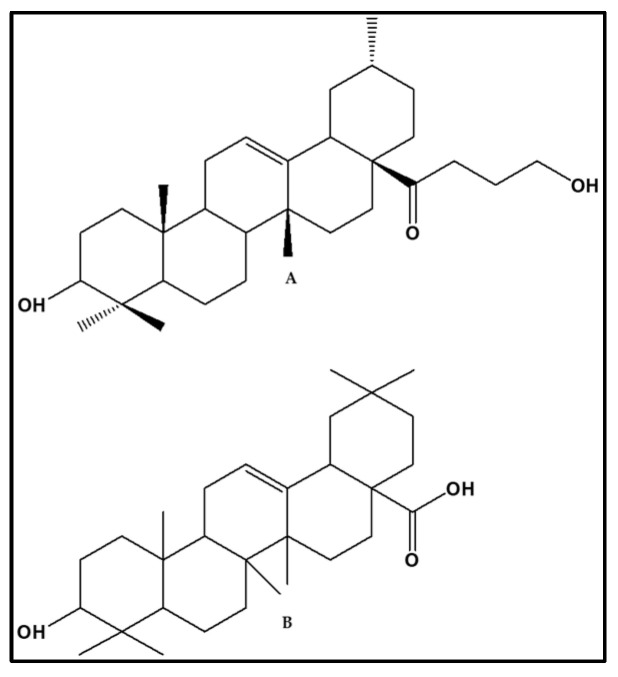
Structure of **D4** elucidated as olean-12-en-28-butanol-β-one-3β-hydroxy-4,4-bis-β,10α, 14α, 20α-pentamethyl (**A**) and its analogue olean-12-en-28-oic, 3-hydroxy-15,16-[1-methylided)bisoxy] (**B**) detected in the dichloromethane extracts by GCxGC-MS and nominated as a major marker compound in South African mistletoe by chemometric analysis.

**Figure 4 plants-13-01382-f004:**
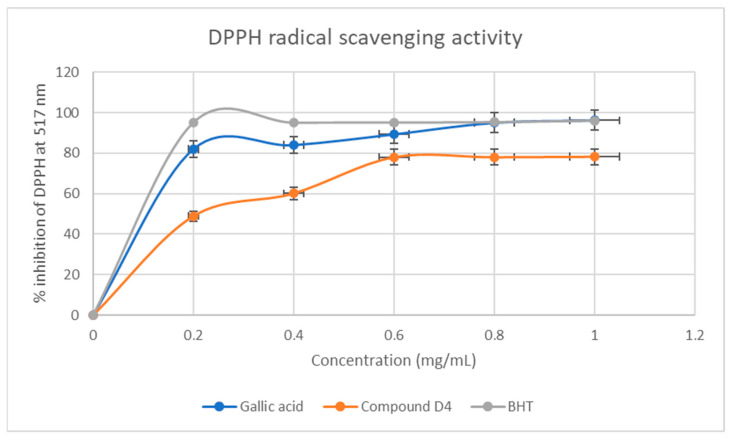
DPPH free radical scavenging percentage inhibition of **D4** and standards. Each value is expressed as mean ± standard deviation of *n* = 3.

**Figure 5 plants-13-01382-f005:**
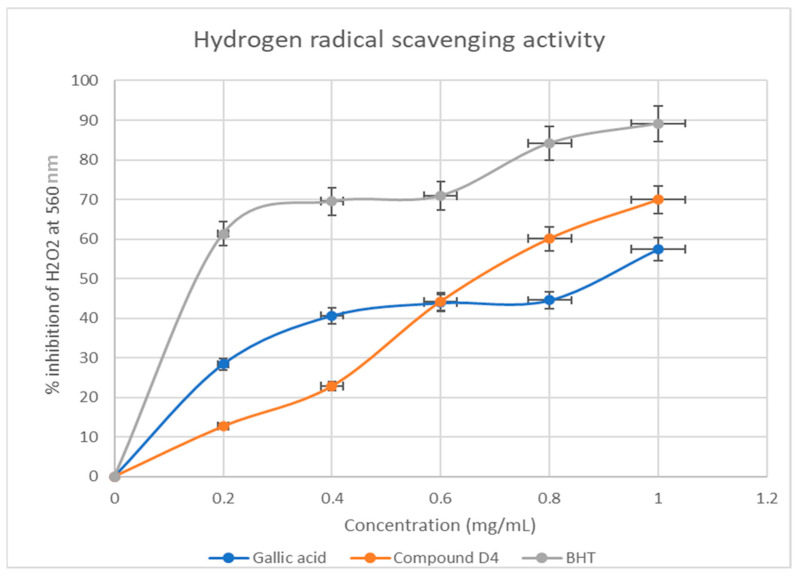
Hydrogen peroxide free radical scavenging percentage inhibition of **D4** and standards. Each value is expressed as mean ± standard deviation of *n* = 3.

**Figure 6 plants-13-01382-f006:**
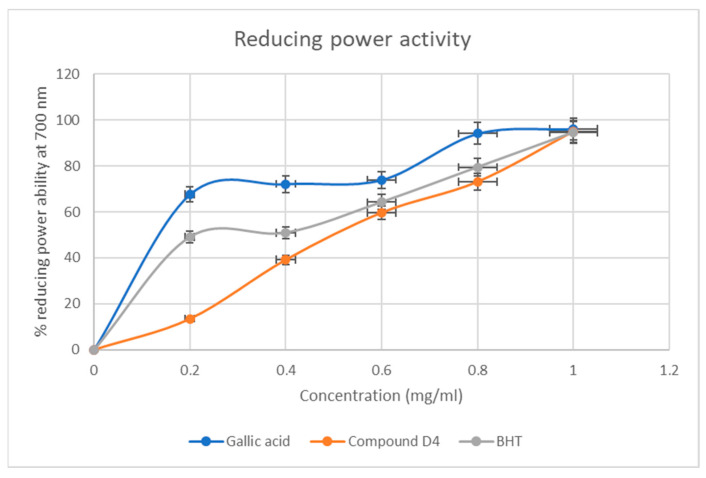
Ferric chloride reducing power activity of **D4** and standards. Each value is expressed as mean ± standard deviation of *n* = 3.

**Table 1 plants-13-01382-t001:** Summarized chemical shifts, multiplicity and coupling constants of **D4**.

Position	13C NMR OF D4	1H (Multiplicity)
1	32.63 (CH_2_)	1.76 (dt, J = 27.4, 13.6, 4.2 Hz
2	27.19 (CH_2_)	1.66 (m)
3	79.05 (CH)	3.24 (dd, J = 11.4, 4.4 Hz
4	46.53 (C_Q_)	-
5	38.41 CH)	0.94 (m)
6	23.40 (CH_2_)	1.89 (m)
7	22.93 (CH_2_)	1.67 (m)
8	45.89 (CH)	1.24 (m)
9	37.10 (CH)	1.41 (m)
10	39.29 (C_Q_)	-
11	25.93 (CH_2_)	1.11 (dt, J = 14.5, 4.90 Hz))
12	122.66 (CH)	5.32 (t, J = 3.7 Hz)
13	143.59 (C_Q_)	-
14	33.81 (C_Q_)	-
15	27.69 (CH_2_)	1.78 (m)
16	23.57 (CH_2_)	1.95 (dt, J = 10.0, 4.7 Hz)
17	41.00 (C_Q_)	2.82 (m)
18	41.61 (CH)	2.88 (m)
19	38.76 (CH_2_)	1.63 (m)
20	32.45 (CH)	1.84 (m)
21	30.67 (CH_2_)	1.30 (m)
22	28.10 (CH_2_)	1.02
23	15.53 (CH_3_)	0.81 (s)
24	15.32 (CH_3_)	0.93 (s)
25	18.30 (CH_3_)	1.59 (s)
26	17.14 (CH_3_)	0.79 (s)
27	1.01 (CH_3_)	0.09 (s)
28	183.14 (C_Q_)	-
29	33.06 (CH_2_)	1.67 (m)
30	47.64 (CH_2_)	1.55 (dt, J = 14.2, 5.3 Hz)
31	55.24 (CH_2_)	0.79 (d, J = 8.9 Hz)

**Table 2 plants-13-01382-t002:** MIC Results for D4 (mg/mL) antimicrobial potentials.

		MIC (mg/mL)			
Analytes	*Pseudomonas aeruginosa*	*Streptococcus pyogenes*	*Staphylococcus aureus*	*Bacillus subtilis*	*Escherichia coli*
**D4**	0.25	0.25	0.25	0.25	0.25
Ciprofloxacin	0.0039	0.0039	0.0078	0.0156	0.0039

**Table 3 plants-13-01382-t003:** IC_50_ values of **D4** and standards indicating inhibitory potentials of **D4**.

IC_50_ Values in mg/mL			
Analytes	DPPH Scavenging	H_2_O_2_ Scavenging	Fe^+3^ Reducing Power
D4	0.398	0.701	0.533
Gallic acid	0.175	0.793	0.284
BHT	0.072	0.329	0.422

**Table 4 plants-13-01382-t004:** The MS, UPLC, and gradient system analysis conditions.

MS Conditions		
Detector	Waters^®^ Synapt G2QTOF	
Calibration mass range	50–1200 *m*/*z*	
Capillary voltage	ESI+ 2.6 KV; ESI− 2.4 KV	
Ionization mode	Both ESI+ and ESI−	
Source temperature	120 °C	
Sampling cone	20 V	
Extraction cone	4.0 V	
Desolvation temperature	300 °C	
Cone gas flow	20.0 L/Hr	
Desolvation gas flow	600.0 L/Hr	
Data management	MassLynx^TM^ (version 4.1 UNIFI)	
**UPLC Conditions**		
System	Waters Acquity UPLC	
Column	Kinetex^®^ 1.7 µm EVO C18 100 Å (2.1 mm ID × 100 mm length)	
Injection volume	5 µl	
Column temperature	50 °C	
Sample temperature	8 °C	
Flow rate	0.3 mL/min	
Mobile phase A	Water + 0.1% formic acid	
Mobile phase B	Acetonitrile 0.1% formic acid	
**Gradient**		
**Time (min)**	**%A**	%B
Initial	97.0	3.0
0.10	97.0	3.0
14.00	0	100.00
16.00	0	100.00
16.50	97.0	3.0
20.00	97.0	3.0

## Data Availability

The data used in the current study are contained within the article and Appendix A.

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
