# Peer review of "Isolation of a Marker Olean-12-en-28-butanol Derivative from *Viscum continuum* E. Mey. Ex Sprague and the Evaluation of Its Antioxidant and Antimicrobial Potentials"

_plants, 2024, doi:10.3390/plants13101382_

Round 1

Reviewer 1 Report

Comments and Suggestions for Authors

The overall research article discuss about the isolation of marker compound 'olean-12-en-28-butanol derivative' from the mistletoe tree (South African ecotype) and its antioxidant and antimicrobial properties. 

- Overall results identify the two compounds D3 and D4 from the mistletoe tree, but only one was identified because the other one was not pure enough. Need more elaboration on this part

- the MIC values fro D4 compound is all same, which does not seem right. need more explanaation

- No supplementary information to support this article

- A lof of sentences do not have any reference to support the claim

- The author said this is a marker compound, but there is no evidence of the extract that says this is the marker compound.

- Discussion section is written as the template, the author did not write anything in this part.

Comments on the Quality of English Language

Overall English editing is majorly required. English grammar is poor and a lot of typos and errors have been found. 

Author Response

Authors rebuttal to reviewer is attached with thanks.

Dr Bassey

Reviewer 2 Report

Comments and Suggestions for Authors

This paper focuses on the isolation and structural determination of compounds from the mistletoe tree. 
The manuscript presents multiple serious flaws which lead me to recommend its rejection. Mainly, the results section is not organized following the standard practice in the field. Spectra should not be added in the body of the paper, but only in the supporting information, and experiments conducted should not be described separately, but as a coherent narrative when describing how the identity of the different compounds was elucidated. The discussion section is missing, the stereochemistry of the compounds is not described, and more importantly, the compound D4 is described as an oleanane triterpenoid but CH3-26 is misplaced for that skeleton (it should be attached to C-8), while CH3-27, which should be alpha oriented and attached to C-14, is missing.

Comments on the Quality of English Language

The manuscript is not of sufficient English quality.

Author Response

The authors rebuttal to the reviewer is attached with thanks.

Dr Bassey

Reviewer 3 Report

Comments and Suggestions for Authors

The submission reports isolating and structure elucidation of the compound (olean-12-en-28-butanol-1-one,3-hydroxy-4,4,10,14,20 pentamethyl) from South African mistletoe tree, followed by providing evidence for its antioxidant and antibacterial activities. Overall, the work is comprehensive and detailed, has good general interest despite the fact that its analogue was previously identified from the same plant species.

The reviewer has some comments and suggestions for the authors to further improve the manuscript:

11)      The submission reports subsequent extraction (Figure 1) with multiple organic reagents, firstly by hexane, followed by DCM, then by Acetone and finally methanol. It is noted that dry extract weight increased from 3.4 g, to 5.1, then 5.2 and 10.5 after methanol extraction. This is contrary to the concept of isolation/purification, i.e. only a subset of compounds will be extracted by a subsequent extraction step. The authors need to explain why subsequent extraction increased dry weight of abstract instead of decreased it.

22)      The official and specific Latin names should be provided for each tree species.  For plant material with no confirmed species name, morphological features including photos should be provided, preferably with a deposit of voucher plant in a public research institution.

33)      Likewise for chemical structures, their unique identifier should be provided, such as CAS (Chemical abstracts service) registry numbers.   

44)      There is the lack of details on later part of isolating extract by column chromatography (section 4.4), like:

aa) ‘dry silica’ if not informative enough, the brand and grade information should be provided.

  b) Not enough details on how the chromatography was conducted like flow speed and volume of collection for each “Test Tube”

55)      Section 3 has no content (lines 240-244), authors should add

66)      In the discussion, it is suggested to include:

a)       Why was the same compound not isolated from other mistletoe trees?

b)      If this compound is the dominant compound in the organic solvent extracts, for this sample and other South African samples.

c)       Why the authors suggest that it could a standard reference for quality control for ‘any’ mistletoe based commercial products.

d)      What could be the potential use for this pure compound? What is toxicity profile if any.

Comments on the Quality of English Language

Quality of English Language is acceptable.     

Author Response

The Authors to the reviewer is attached with thanks.

Dr Bassey

Reviewer 4 Report

Comments and Suggestions for Authors

In present manuscript, authors aimed to isolate a compound from the South African mistletoe extracts and to evaluate its antioxidant and antimicrobial potentials as a standard reference for quality control of any mistletoe based commercialised products. In generally, the manuscript was well prepared; methods were clear. However, some key points should be addressed before acceptance.

1.     Figure 1, I suggest author delete this image. The data can be expressed using table.

2.     Figure 2, Figure 3, I suggest this figure can be replaced high solution image.

3.     For MIC Results for D4, I suggest the images should be added in text.

4.     Figure 6, Figure 7, and Figure 8, why the SD were expressed different type? Why Figure 8 no SD?

5.     The required references should be cited to part M & M

6.     Materials and methods: following all the reagents and equipment, the information of company, city, country should be included.

7.     References list only ten references. For scientific paper, this is impossible. Thus, I suggest author must update the references.

8.     The writing English should be improved by native English speaker. I strongly suggest to check the manuscript carefully, especially please check the grammar and the completeness of the sentences once again. And please check the tense of the sentences. There should be consistency throughout the manuscript using past tense.

Comments on the Quality of English Language

In present manuscript, authors aimed to isolate a compound from the South African mistletoe extracts and to evaluate its antioxidant and antimicrobial potentials as a standard reference for quality control of any mistletoe based commercialised products. In generally, the manuscript was well prepared; methods were clear. However, some key points should be addressed before acceptance.

1.     Figure 1, I suggest author delete this image. The data can be expressed using table.

2.     Figure 2, Figure 3, I suggest this figure can be replaced high solution image.

3.     For MIC Results for D4, I suggest the images should be added in text.

4.     Figure 6, Figure 7, and Figure 8, why the SD were expressed different type? Why Figure 8 no SD?

5.     The required references should be cited to part M & M

6.     Materials and methods: following all the reagents and equipment, the information of company, city, country should be included.

7.     References list only ten references. For scientific paper, this is impossible. Thus, I suggest author must update the references.

8.     The writing English should be improved by native English speaker. I strongly suggest to check the manuscript carefully, especially please check the grammar and the completeness of the sentences once again. And please check the tense of the sentences. There should be consistency throughout the manuscript using past tense.

Author Response

The Authors rebuttal to the Reviewer is attached with thanks.

Dr Bassey

Reviewer 5 Report

Comments and Suggestions for Authors

The manuscript entitled  “Isolation of a Marker Olean-12-en-28-butanol Derivative From Mistletoe Tree and the Evaluation of Its Antioxidant and Anti-Microbial Potentials aims to isolate a compound from the South African mistletoe extracts and to evaluate its antioxidant and antimicrobial potentials as a standard reference for quality control of any mistletoe based commercialized products.

Line 33, 34. Authors should combine and rewrite 2 sentences in one.

Coupled with the many traditional uses, the phytoconstituents in the plant are known to be responsible for the biological activities and such uses. Remove this sentence or connect with first, because authors introduced us with Mistletoe tree not with general medicinal herbs.

Line 45. -O- should be italic.

Authors should add paragraph about biological activities (antimicrobial,…) of isolated compounds and plants with more details. It is important because of potential application of Mistletoe tree.

Line 102 m\z should be italic.

Figure 3. Resolution of Figure is too low. Authors should improve it.

Authors should add part about discussion of results and compare it with other studies.

Author Response

The Authors rebuttal to the Author suggestion is attached with Thanks.

Dr Bassey

Round 2

Reviewer 1 Report

Comments and Suggestions for Authors

- Figure 5 and 6 there needs to be a space between the nm

- The quality of figure 2 and 3 can be drawn better. There are some size differentiation

- for Table 1, it needs to be 13C and 1H instead of C13 and H-1

- Line 108, 'd4' needs to be italicized

- The author said negative ion mode, but the repsentation is [M+H]-. Correct it.

- Line 103, purity 96.2% not 96,2%. In addition, purity based on what? UV area? the supplementary figure S1 does not show the area value

- Line 202 and 213, n number is not shown

- For Table 3, shouldn't the experiment be in triplicate? if so, why is there no representation of standard deviation?

- The conclusion lack the motif of the study

Comments on the Quality of English Language

Minor english revision required

Author Response

We appreciate the input from the this reviewer

Reviewer 3 Report

Comments and Suggestions for Authors

The authors have addressed my comments and suggestions satisfactorily, I appreciate your diligence.

Author Response

The contribution from this reviewer is highly appreciated

Reviewer 4 Report

Comments and Suggestions for Authors

The manuscript was improved enough, I suggest accept it with present version.

Author Response

We appreciate the suggestions from this Reviewer with many thanks

Reviewer 5 Report

Comments and Suggestions for Authors

Authors improved manuscript.

Authos could improve sentance. Line 63.

Here, there is one suggestion.

Despite an extensive literature search, no compounds isolated from the South African mistletoe tree have been documented to date.

Author Response

(The authors gave the same response as above.)
